# What Is an Extreme Sports Healthcare Provider: An Auto-Ethnographic Study of the Development of an Extreme Sports Medicine Training Program

**DOI:** 10.3390/ijerph19148286

**Published:** 2022-07-07

**Authors:** Larissa Trease, Edi Albert, Glenn Singleman, Eric Brymer

**Affiliations:** 1Healthcare in Remote and Extreme Environments (HREE) Program, School of Medicine, University of Tasmania, Hobart, TAS 7005, Australia; edi.albert@utas.edu.au (E.A.); glenn.singleman@utas.edu.au (G.S.); 2La Trobe Sport and Exercise Medicine Research Centre (LASEM), La Trobe University, Bundoora, Melbourne, VIC 3086, Australia; 3Faculty of Health, Southern Cross University, Gold Coast Campus, Bilinga, Gold Coast, QLD 4225, Australia; eric.brymer@scu.edu.au

**Keywords:** extreme sports medicine, medical education, injury, extreme sport athlete

## Abstract

“I remember when sex was safe and skydiving was dangerous” read a popular bumper sticker during the HIV crisis. Popular perceptions of extreme sport (ES) often include the descriptor ‘dangerous’. Therefore, why is the popularity of ES increasing exponentially with “dedicated TV channels, internet sites, high-rating competitions, and high-profile sponsors drawing more participants”? More importantly, how should health practitioners respond to the influx of ES athletes with novel injuries, enquiries and attitudes. This paper describes the results of a collaborative auto-ethnographic approach to answering “what is an extreme sports medicine health care provider and what are the components of an effective Extreme Sports Medicine (ESM) training program?” The study was conducted following the first ESM university course offered in Australia with the intention of assessing the learning design and reflecting on the development and practice of ES health practitioners. We explicated three overarching themes common to both the ES health practitioner and for the effective training of healthcare providers in the support of ES endeavors and athletes. These themes were individual, task and environmental factors. The impacts of these findings confirm that ESM courses are vital and should be designed specifically to ensure that practitioners are effectively supported to develop the unique skills necessary for practice in real world extreme sports events.

## 1. Introduction

Extreme sports have erupted onto the sporting scene over the last decade [1,2,3,4], with several new sports appearing [5,6] and the boundaries of traditional ones being radically redefined [7]. While many extreme sports are practiced as individual adventure activities, others have become organised, competitive, and occasionally commercial sports. Dedicated events like the X-Games and World BASE Race host extreme sport athletes from around the world and are watched by hundreds of millions on digital media platforms. Some sports have been mainstreamed with inclusion into the Olympic Games calendar: sport climbing, free ride BMX and surfing (Tokyo 2020) and ski mountaineering (Innsbruck Youth Olympic Games, 2020 and Milano Cortina, 2026). In the ‘click bait’ world of digital media, some extreme sport athletes have become superstars and popular culture has transformed athletes once regarded as the ‘lunatic fringe’ into ‘Supermen’ [8].

Extreme sports have been defined as “independent adventure activity where a mismanaged mistake or accident is most likely to result in death” [9]. In sporting events this definition has been expanded to allow for emerging and competitive activities, in these instances the notion of “action sports” has been used to frame a broader perspective on modern extreme sports. The action sports definition then permits the inclusion of sports such as cliff diving, sport climbing, and foiling. Health practitioners working in this area need to not only understand the athletes and sports more commonly encountered, but also require a framework with strong foundations in risk management, individual athlete assessment and event planning and delivery to allow them to explore opportunities to provide care in any extreme sporting or adventurous endeavor and environment that captures their interest.

As the levels of participation within the community, and the push for increased performance amongst elite extreme sports athletes has evolved, the medical care of these athletes has begun to catch up and a new medical sub-discipline of extreme sports medicine (ESM) has been born [10]. Unsurprisingly, the leaders in this field are predominantly from Europe [11] and North America [12], but a formal university healthcare program broadening the traditional medical teaching horizons to include BASE jumping, paragliding, mountaineering, extreme skiing, skydiving, breath hold diving and ultra-endurance running, etc. has been lacking.

This paper describes an auto-ethnographic approach to answering “what is an extreme sports medicine health practitioner and how can they be trained” based on the reflections of the developers of the Extreme Sports Medicine unit that forms a part of the Certificate, Diploma or Masters in Healthcare in Remote and Extreme Environments (HREE) program through the School of Medicine at the University of Tasmania, Australia. We examine the evolution of ESM and report on findings from an ESM training program with the dual aims of answering the question “what is an ESM health practitioner” and identifying the challenges and opportunities for developing effective ESM training programs.

## 2. Sport and Exercise Medicine

Arguably ESM is most directly related to Sport & Exercise Medicine (SEM) which is currently a recognised medical specialty in sixteen European countries with a training period that varies between two and five years [13]. The General Medical Council (GMC) training curriculum [14] for UK SEM recognises the SEM Physician as a leader of a multi-disciplinary team in the management of musculoskeletal conditions, including the care of elite athletes. Despite the wide acceptance of SEM as a specialty and the recognised role of the SEM Physician in athlete care, there is a relative lack of teaching of SEM skills at medical schools [15,16,17,18,19,20], and in an already minimally taught specialty, extreme sports medicine is a single dot point in an extensive curriculum [21].

While ESM and SEM converge in the management of athletes’ health and performance, they diverge with respect to the challenge that the task and environment may present to both athlete and health practitioner and the subsequent impacts on psychology, technology and risk management. A concept explored in depth by Buckley [22] who described the difference in psychology between adventurous and extreme sports. With constrained teaching within the traditional medical school and post-graduate specialty training programs, ESM has evolved through health practitioner lead research, educational opportunities and overlap with established programs in Wilderness and Expedition Medicine [23,24,25]. Hoffman and colleagues have published extensively on medical coverage considerations for ultra-endurance events [26,27,28,29] and offer an online training program for medical providers, event organisers and coaches in ultramarathon medical care [30].

While there are a wide range of sources offering information and ideas related to training and psychology in various sports, texts in ESM are still few and far between, although Schoeffl et al. [31] first provided a text on rock climbing injuries as far back as 2003. More recently Feletti edited the first comprehensive Extreme Sports Medicine text [10]. Enthusiasts are also able to upskill in ESM at conferences in Europe (Feletti, UTMB) and North America (Colorado). These conferences, however, are not a prescribed syllabus, nor a framework for learning for students of ESM.

## 3. The Role of Medical Practitioners in Extreme Sport Events

Most organised Extreme Sport Events like the X-Games, World BASE Race and ultramarathons have doctors in attendance at the finish line. The increased participation rates, number of events, bigger cross section of participants (some with impairments and medical issues) and access to extreme sports more generally has also meant an increase in the desire for knowledge on the prevention and treatment of injuries specific to extreme sport activities [32]. Feletti and colleagues have described injury patterns specific to both dinghy-sailing on hydro-foiling boats [33] and kite buggying [34]. In addition to a focus on sport specific injuries, the University of Tasmania Extreme Sports Medicine course examines in detail medical presentations in ES athletes, including exercise associated hyponatraemia, exercise associated collapse, hyperthermia, hypothermia, frostbite, shallow water black-out, ‘samba’, high altitude pulmonary edema, high altitude cerebral edema, high altitude retinal hemorrhage, acute altitude related hypoxia, altitude related barotrauma and decompression sickness. These novel injury and illness problems require health practitioners with experience to manage on-site issues and work with athletes and event organisers prior to events to optimise planning and reduce risk. Most extreme sport event health care providers have an interest in the relevant sport but few have formal, extreme sport specific training. Event organisers have no way of deciding what kind of medic should be employed for ‘first aid support’ or ‘risk management advice’ and worse still, athletes do not know what health care providers they can visit for empathetic, non-judgemental information about physical training, performance, diet, risk management and mental preparation for extreme sport. Healthcare providers studying extreme sports medicine have potential to be an asset to the health of the broader community as they are more likely to prescribe their patients physical activity in nature and expand the sporting activities they can prescribe, potentially leading to greater patient engagement, especially amongst younger community members [35].

To support extreme athletes to optimal health, safety and performance, healthcare providers need to be up skilled in training methodology, nutrition, injury and illness prevention, engineering and safety equipment, management of the injured athletes and medical event coverage specific to extreme sports. Extreme sports medicine health practitioners must not only understand the physical demands and injuries of extreme athletes but the psychology of the sports, the process of risk assessment and mitigation and the role of cognitive biases, all of which can influence outcomes for the ES athlete [36]. This suggests that the training of extreme sport practitioners needs to move beyond the technical development and injury management focus of sport medicine courses into wider, sometimes novel, domains. Further training courses that aim to develop highly skilled ESM practitioners need to be carefully designed with learning that is more representative of the practitioner and task requirements, and environment context which is often practiced in remote and austere environments more similar to wilderness medicine [37,38].

## 4. Learning Design in Extreme Sport Medicine

Education and learning design in the field of medicine has been critiqued for relying on the traditional idea of teaching as an intervention that somehow results in achieving the specified learning outcomes [39]. According to Biesta and Braak [39] this model stems from the ‘medical model’ prevalent in medicine and does not do justice to how people learn. As an ESM health practitioner will be required to perform in extreme environments, learning design needs to reflect a more up to date context. As an ESM health practitioner will be required to perform a considerably expanded role compared to conventional medicine, not only does the learning design need to reflect a more up to date approach to medical education, but also take account of the specific contexts of supporting extreme sports event and endeavors. As this is the first university course to provide scope for learning ESM skills with both a sporting performance and athlete health focus, leaning design was an important consideration. The course explores the physical demands and injuries of extreme athletes as well as the psychology of the sports, the technology of the sports, the process of risk assessment and mitigation and the role of cognitive biases. The course is designed to sit within a program of healthcare in remote and extreme environments and the focus is on caring for athletes who are undertaking their endeavors in extreme environments. The program has two key aims which, while focused on the extreme sport athlete or event, provides a template to facilitate healthcare practitioners providing care to active individuals either in a one-on-one clinical setting or through event coverage. The course design team has had previous experience designing novel and non-standard courses [40] and was cognisant of designing the structure and content to both fit into an existing program as an elective unit [25] and to be self-contained as a standalone course.

The over-arching structure for the curriculum and the assessments was provided by Miller’s pyramid as a means to create a progression from knowledge through to application of knowledge and translation into clinical practice [41]. Within this hierarchy, adult learning theories were examined and those related to instrumental learning (cognitive and experiential) and social learning in particular, provided a foundation for the course design [42].

An Online learning approach [43] was necessary given the geographical spread of prospective students, although the limitations to this were appreciated and elements of the online material were extended to allow teaching of practical skills through a residential camp.

The Delphi technique is well recognised and has been used to create curricula for conventional sports medicine [21,44]. A modified Delphi process was used for UTAS ESM at two levels. Firstly, the core academic group used it to create the intended learning outcomes for the course and to create a common structure to write content for each sport (Table 1). The latter was done for several reasons. Firstly, it provided structural familiarity as students moved through the modules, secondly it was intended to help content experts contribute more effectively and consistently. Finally, it was hoped that this structure could be used for students to examine additional sports themselves (that were not included in the course). The individual sport modules were delivered following three “common themes” modules which established a base level of knowledge in areas that transected all sports, including the science of performance, nutrition, psychology, risk management, the environment, medico-legal considerations, and anti-doping as they relate to extreme sports.

The second level was to create and review the content at the level of the individual modules from multiple sources including the core academic group, additional colleagues from within the Healthcare in Remote and Extreme Environments Program and external content experts who agreed to collaborate with us.

## 5. Collaborative Autoethnography Methodology

The aims of our study were to examine the questions of “what is an ES health practitioner” and “how can they effectively be trained.” In this paper we followed a collaborative autoethnography approach on the first extreme sport medicine course offered in Australia with the intention of identifying the characteristics of an ES health practitioner and assessing the learning design and impact of our inaugural ESM course in meeting the needs of a developing or novice ES health practitioner. Whereas autoethnography typically involves an individual researcher deliberately utilising a retrospective process to make explicit personal stories in a cultural context, collaborative autoethnography involves multiple researchers [45]. Autoethnography combines intentional and considered analysis [46] of biographical accounts and personal experience to bring to life cultural experience. Collaborative autoethnography is ideally suited to stories that provoke collaboration [47]. Collaborative autoethnography provides the opportunity for ‘collective interpretation’ [45] and the examination of inherent assumptions and presuppositions. It has been described as an effective methodology for advancing the understanding of the culture and practice of medical education [48,49].

In this paper, we have used a collaborative process to explicate and interpret the experiences of the first three authors reflection on the qualities of an ES health practitioner and the development and implementation of the first ESM training program. Trease, Albert and Singleman conducted an independent written reflection on their experience as ES health practitioners and in designing and facilitating Australasia’s first postgraduate extreme sport medicine course designed to enhance the capacities of medical professionals to support extreme sport endeavors and events. Brymer, undertook the initial analysis by reading the reflections multiple times before commencing a thematic analysis process. A categorical-content perspective [50] was utilised first, where the text was first broken down to self-contained areas of content. The thematic analysis process then followed the recommendations outlined by Braun & Clarke [51] for generating initial codes, searching for themes, reviewing themes, defining and naming themes, and producing a summary table of themes and direct quotes. Subsequently all authors explicated the experience in an attempt to draw meaning that considered the current theoretical frames of references and interpreted the experience in light of current cultural perspectives. Brymer acted as critical collaborator [9] helping to structure the learning from Trease, Albert and Singleman’s stories for an academic and practitioner audience. This article provides an analysis of these accounts. The experience is interpreted partially from the perspective of research into learning design in medicine.

## 6. Results

Three overarching themes emerged from our study examining the question of “what is an ESM health practitioner” and identifying the challenges and opportunities for developing effective ESM training programs for healthcare providers. The three themes were: Individual (learner) factors, Task factors and Environmental factors (Appendix A).

### 6.1. Individual Factors

Extreme sport medicine courses must have a strong component of learner centered relevancy. This is not just in the sense that curriculum needed to be delivered in a manner that suited individual learners but also that the learning design needed to facilitate the development of adaptable, multi-skilled and ecologically minded professionals. For example, EA reflected that an effective ESM health practitioner had multiple hats that were not necessarily related to traditional medicine or sport medicine training and that ESM was unlikely to be their sole career choice. Health practitioners are likely to bring skills from other aspects of their professional life to the ESM training context. Health practitioners could come from multiple related medical backgrounds as there was, in the words of one of the authors (EA):


*“no single ‘fit’ but rather a health practitioner (and certainly not necessarily a doctor) who has the qualities of a sense of humility and discovery and who can understand the environment, understand the sport and the equipment and technology that accompanies it, understand the athlete and be able to work pro-actively as well as reactively in a setting pertinent to that sport.”*


EA went on to reflect that a one-size-fits all approach to learning design would not work for ESM courses as in many cases learners are not sport experts or even recreationalists, instead they need to have a broad knowledge and a way of being able to fit that knowledge into contexts relevant to the ESM practitioner. This aligned with the reflections of another author (LT):


*“No student, nor faculty member, nor ESM health practitioner will be an expert in all of the sports, so the course needed to allow students to do a “deep dive” (pun intended) in two or three sports but offer enough sports that every student could choose a diverse range of content to support their learning.”*


One of the authors (GS) also reflected how his experience had taken him across the world in different extreme environments and different sports. He also (GS) reflected:


*“I have come to realise that there won’t be a single type of ESM health practitioner and we don’t need that to enhance the health and performance of extreme sports athletes. What we need is practitioners from multiple disciplines, who are interested and knowledgeable in a (or more than one) sport, with the skills to help the athlete or event organiser, in the way that they will be working.”*


Beyond the immediate impact, one of the authors (LT) expanded further reflecting that course design needed to provide a framework for lifelong learning or professional development as ESM practitioners are faced with personal challenges when participating in extreme contexts as well as problem solving medical challenges that are often specific to particular events.

### 6.2. Task Factors

Perhaps as would be expected in any medical training context, there is also a large focus on task related factors relevant to extreme sports. For example, injury types and aspects of risk management. However, what became clear through the course was, unlike more traditional medical training, the context of ESM required a broad set of skills, not just providing treatment for poor outcomes, when something has gone wrong. The nature of extreme sports events and the potential for death or serious injury indicates that the ESM health practitioner needs to have the personal and technical skills to be involved early and provide advice before an event takes place.

The fact that the definition of extreme sport mandates that the practice of the sport must involve risk of severe injury or death establishes separate physical and psychological domains that team sport health practitioners may not be familiar with. An ‘off-day’ or ‘choke’ or ‘error of judgement’ will have different consequences for the ES athlete compared to the golfer, tennis player, cricketer or netballer.

ESM health practitioners need to be part of the planning team, to get to know the athlete(s) and advise as required. Effective ESM support might involve psychologists, engineers and nutrition experts. This was also apparent in the backgrounds of the learners in this course who heralded from diverse medical backgrounds. “Effective ESM training needs to be able to provide a meaningful space for a variety of practitioners to gain the skills required to effectively support the athlete working across the ‘proactive, preventative or performance space” (LT). One of the authors (GS) summarized this as “needing to work beyond the standard medical model.”

### 6.3. Environment Factors

The main environmental factor identified in our autoenthnographic work was the development of a culture of passion and enthusiasm for “becoming a scholar of extreme sport”. This was complimented by the need for the acceptance of a wide range of expertise in ES healthcare providers, which may include experiential, rather than formal learning. Interestingly while the physical environment obviously plays an import part in participation or healthcare provision, it seemed to be less relevant in training ES healthcare providers. Instead the emphasis was on ensuring the atmosphere or culture of the course encouraged students to adopt a broader perspective on ESM.

That is, the teaching team was very important in the overall training context. Not just because of their expertise but also because of the way they approached the topic. As described by one of the authors (EA):


*“in some extreme sports, there was a real need to seek out the nuanced expertise that couldn’t be gleaned from books or papers, if we wanted our students to have an authentic learning experience.”*


This meant that to create the ideal ESM faculty, EA described:


*“you need people who have the depth of clinical experience, a deep understanding of preferably several extreme sports and the interest in education, and somehow be able to employ and stimulate these ‘out of the box’ non-conformist souls within a very ‘closed box’ conformist University setting.”*


## 7. Discussion

Learning design in medicine has traditionally followed the path of discipline orientated specialisation (Sports Radiologist or Sports Orthopaedic Surgeon). Critics of this approach have pointed to the need for learning design that better fits the individual learner and their desired work environment and focus. This study indicates that effective learning design in ESM courses is not about creating experts with narrower specialist skills but instead facilitating a lifelong learning journey that broadens the learner’s skill set, beyond that of their individual discipline. This approach can enable the ES health practitioner to pursue knowledge of an emerging ES sport, injury, technology or treatment and the importance of teaching this skill of lifelong learning is well documented in both medical [52] and non-medical education literature [53,54].

As learners will come from various backgrounds, both within medicine and broader healthcare roles, any ESM course should provide opportunities for each leaner to design their own professional journey. Our findings also show that ESM training courses need to go beyond simply designing learner centred experiences [42] to develop a learning environment that facilitates passion for ES, the environment in which they are practiced and the values and belief systems of the ES participant. Our study identifies the importance of the faculty in creating an environment where learners are given permission and challenged to think ‘out of the box’, just as many ES athletes do [55]!

In contrast to traditional sport medicine [15,19] the skills required by an effective ES practitioner are arguably more aligned with those required for effective wilderness medicine. These skills include the capacity to not only understand the environment, often austere, that ES are ‘played’ in [55], but to be able to effectively deliver care in such an environment [38], while ensuring their own personal safety and well-being, a factor which is infrequently a consideration when providing team sports coverage on a pitch or court [21]. The implications from our study in the theme of individual factors for ESM education are quite profound in that learners need to be prepared to work effectively in unknown and austere environments where individuals might feel uncomfortable [39]. ESM training needs to effectively prepare the individual for this personal challenge, whilst equipping them with the professional skills required.

ESM health practitioners can be called on to provide both health and performance advice proactively as well as healthcare support before, during and after an event, but maybe challenged by an inability to offset the significant risk of injury or death during the endeavor [39]. In the theme of task factors in medical training focused on extreme sports, we identified the need to develop skills for identification of risk, implementation of effective prevention strategies, as well as the treatment of conditions arising during participation. The ESM health practitioner needs skills that are beyond their singular discipline, that is rather than becoming narrower through specialist extreme sports medicine training. Similar to wilderness medicine [38] the education experience needs to encourage learners to adopt a broader set of skills [39].

Extreme sports are emerging global activities that are rapidly out performing more traditional sporting activities [55]. An important impact of this is the rising need for well qualified, experienced health practitioners capable of practicing their skills in extreme environments with uncertain outcomes [39]. While there is some similarity to the development of effective wilderness medicine practitioners [38] there are unique requirements for extreme sports medicine. Coupled with the broad skill set outlined above and the capacity to react to multiple medical scenarios unique to particular sports and the environment in which they are practiced ESM practitioners need to provide care across the spectrum of prevention to rehabilitation for both the body and mind.

The implications from this study are that specific ESM practitioners and courses that train them are vital and should be encouraged. However, following a traditional medical approach to course design is not ideal for ESM training. While there is cross over with wilderness medicine practice there are also differences. ESM course designers need specialist skills to ensure that learners are effectively supported for real world extreme sport events. This study is the first to provide guidance on what that entails and principles for effective ESM course design.

## 8. Conclusions

In our autoenthnographical analysis of the question “what is an extreme sports medicine health practitioner” we identified that the ESM health practitioner is able to go beyond the question of ‘why would you do that?’ and into the territory of ‘how can I help you manage the physical, emotional and psychological risk of that extreme activity?’ The ESM health practitioner understands the physical risks of the sport and the psychological motivations of the athlete. The ESM health practitioner can offer non-judgemental advice on how to maximise physical preparation and flawless execution while staying within the bounds of “acceptable risk”. The ESM health practitioner can help an athlete foster and maintain psychological balance in extreme environments and extreme challenges. The ESM health practitioner can help identify and mitigate cognitive bias and unacceptable risk. The ESM health practitioner can be an objective, voice of reason more easily accepted and trusted by the ES athlete. Finally, the ESM health practitioner can be equipped with the most appropriate medication, gear and skills to provide treatment/stabilization/evacuation to the injured ES athlete at events or individual challenges. Our study highlights the importance of an ESM curriculum which instills these qualities into aspiring ESM health practitioners through a challenging and supportive environment, created by a skilled and divergent faculty with a focus on empowerment towards lifelong learning and the pursuit of a broad skill-set.

## Figures and Tables

**Table 1 ijerph-19-08286-t001:** Template for individual sports and rationale for inclusion in course.

Introduction to the sport	The introduction aims to provide exposure to the sport through open access video and blog posts of extreme endeavours and events.
Legends and pioneers	An essential component of understanding a sport is to be a scholar of those who have developed and advanced the sport. Health Practitioners will need to speak the language of the athletes they work with and understand the key figures in the sport.
Environment	An understanding of the environment in which the sport is conducted assists evaluation of environmental stressors in preparation, injury and illness.
Risk assessment, cognitive bias and psychology	A case-based learning section on the role of cognitive bias in risk assessment and mitigation in the sport
Equipment	An overview of the typical equipment required for the sport and its role as a contributor to outcome (especially morbidity and mortality).
Athlete types	An understanding of the biomechanical context of the sport and how athlete form, psychological influences and cultural influences within a sport will aid appropriate Health Practitioner assessment (e.g., the ape factor of a climber)
Training routines and modalities	While a broad overview of the science and principles of training is provided in the introduction, a further exploration of training types specific to each sport is included in the template.
Common injury patterns, injury management and rehabilitation	The quality and prevalence of injury surveillance and prevention programs in extreme sports is lacking (in comparison to team and Olympic disciplines) but a review of common injury patterns and management are provided for each sport.
Sport specific screening	Musculoskeletal and medical screening can identify areas for improvement in performance or reduction in risk.
Nutrition	Fuelling and hydration strategies can be specific to the event or sport an athlete is undertaking and this is reviewed in each section (e.g., ad libitum drinking vs. programmed hydration during an ultramarathon)

## Data Availability

Data from this study is provided in the Appendix A. Individual author reflections are available on request.

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
