# Peer review of "What Is an Extreme Sports Healthcare Provider: An Auto-Ethnographic Study of the Development of an Extreme Sports Medicine Training Program"

_ijerph, 2022, doi:10.3390/ijerph19148286_

Round 1

Reviewer 1 Report

A PDF document is attached.

Author Response

Reviewer 1 - Review Report

The manuscript has been thoroughly reviewed by the authors and has been substantially improved in its structure and content, now becoming more enriched.

However, I have a few small comments for the authors, which I specify below.

Lines 112-113:

Since these three pathologies: high-altitude pulmonary edema (HAPE), high-altitude

cerebral edema (HACE), high-altitude retinal hemorrhage (HARH) are not mentioned

again throughout the text, it is not necessary to include their respective acronyms.

Therefore, these 3 respective acronyms can be removed.

Response: we have deleted these three acronyms as suggested

Lines 450-455 (reference nº1); Lines 456-459 (reference nº2); Lines 474-486 (reference

nº11); Lines 495-502 (reference nº16); Lines 537-539 (reference nº34); Lines 545-546

(reference nº38): Different articles (total: 20 papers) are grouped in these 6 references. I think they should  be referenced separately. I consider that it would be a much more optimal way of  presenting references and also more useful for readers. In that case, all the reference  numbering that appears throughout the entire manuscript should be properly adjusted. Line 451:

Finally, in the list of references, some journals appear with their abbreviated name and

others do not. Likewise, some article titles appear with the initials of each word in

capital letters, and others do not. In some references the edition number appears after

the volume number, and in others it does not. In some reference the page number/s seem

to be missing. My advice is that, please, the same criteria be unified for the entire

bibliographic list; so that it has a more according to international standards, and/or the

editorial standards of the journal

Response: we have had a frustrating time with the reference style for this journal and getting the references correct, hopefully this has now been addressed and all reference numbers and titles have been edited, both in Endnote and finally manually within the manuscript.

Reviewer 2 Report

I thank the authors for addressing the majority of my concerns. The article is now much more coherent and the purpose of the paper is aligned throughout to the description of the autoenthnographic approach and the findings presented.  However, the discussion still lacks critical interpretation with minimal supporting evidence. Specifically in the second, third, and fourth paragraphs the authors make interesting statements but without a single comparison with any other academic text or information. The authors need to consider their explanations in light of published educational and medical practices. This reduces the academic quality and strength of the work and still needs addressing

I am unsure if this is a issue with the style of the journal. Usually with references each number is assigned to a single paper/text rather than a group of articles

Author Response

Reviewer 2 - Review Report

I thank the authors for addressing the majority of my concerns. The article is now much more coherent and the purpose of the paper is aligned throughout to the description of the autoenthnographic approach and the findings presented.

However, the discussion still lacks critical interpretation with minimal supporting evidence. Specifically in the second, third, and fourth paragraphs the authors make interesting statements but without a single comparison with any other academic text or information. The authors need to consider their explanations in light of published educational and medical practices. This reduces the academic quality and strength of the work and still needs addressing.

Response: We have included six references in the paragraphs outlined contrasting to other academic texts and publications.

I am unsure if this is a issue with the style of the journal. Usually with references each number is assigned to a single paper/text rather than a group of articles

Response: we have had a frustrating time with the reference style for this journal and getting the references correct, hopefully this has now been addressed and all reference numbers and titles have been edited, both in Endnote and finally manually within the manuscript.

Reviewer 3 Report

Dear Autors,

Although the article has been improved in many places it still does not meet the requirements of scientific articles. The methods used do not give measurable results. The statement cannot be confirmed: The impacts of these findings confirm that ESM courses are vital and should be designed specifically to ensure that practitioners are effectively supported to develop the unique skills necessary for practice in real-world extreme sports events.

Its aim is to convince doctors of the necessity to change the teaching of doctors. However, it does not contribute much to this regard. Such a need is obvious in the case of specialization and already arises in the case of treating professional athletes. Additionally, the article still does not follow the pattern typical of scientific publications. There is no description of the material and methods, no clear presentation of the results, and no statistics, or conclusions. However, it can certainly be used as an application to the directors of medical schools or the ministers of health to introduce changes in teaching.

Kind regards.

Author Response

Reviewer 3 - Review Report

Dear Autors,

Although the article has been improved in many places it still does not meet the requirements of scientific articles. The methods used do not give measurable results. The statement cannot be confirmed: The impacts of these findings confirm that ESM courses are vital and should be designed specifically to ensure that practitioners are effectively supported to develop the unique skills necessary for practice in real-world extreme sports events.

Its aim is to convince doctors of the necessity to change the teaching of doctors. However, it does not contribute much to this regard. Such a need is obvious in the case of specialization and already arises in the case of treating professional athletes. Additionally, the article still does not follow the pattern typical of scientific publications. There is no description of the material and methods, no clear presentation of the results, and no statistics, or conclusions. However, it can certainly be used as an application to the directors of medical schools or the ministers of health to introduce changes in teaching.

Response: Reviewer 3 seems to conflate science with quantitative methodology and is at odds with reviewer 1 and 2 above.  Throughout the review process, reviewer 3 has provided only broad suggestions rather than specific feedback and this has been focused on the lack of quantitative methodology, which given the qualitative nature of this research has not been able to be addressed.

This manuscript is a resubmission of an earlier submission. The following is a list of the peer review reports and author responses from that submission.

Round 1

Reviewer 1 Report

A PDF document containing "General and Specific Comments" to the authors is attached.

Reviewer 2 Report

The topic examined in the paper is novel and interesting and provides an insight into both the discipline of extreme sports and the specific medical and healthcare requirements of the discipline.  The paper is well written and the use of autoethnography provides an alternative but appropriate exploration of the topic.  However, there seems to be a discord within the paper that needs addressing before acceptance. Part of the paper describes extreme sports medical practitioners and the role they play in supporting athletes and event, while another part describes the extreme sports medicine course and how this is a creates a more rounded educational provision rather than the structured teaching often presented in standard medicine courses. Most importantly, these do not match up within the paper, for example the methods and results relate to the latter whilst the discussion and conclusion focuses on the former, and at times is actually contradictory to what was presented in the results section. The authors need to provide an expanded discussion and conclusion that focuses on the autoethnographic reflections of the ESM course and how this relates to or enhances the evidence base of teaching and learning theory and delivery in medicine and sports medicine.

L14 – Citations should not be included in the abstract. If they are really necessary then they should be cited with details rather than numbers as readers will not always have access to the full paper and therefore reference list.

L83 – This section and paragraph provides a clear explanation of extreme sports but to be useful it needs to be earlier and should be part of the introduction

L99 – This is a broad claim with little evidence presented and should be addressed

L102 – The sentence beginning ‘Other ESM related medical issues’ does not adequately follow on from the preceding text, which loosely includes injury as an ESM issue, whereas the following list is very detailed. A clearer bridge between these two statements is necessary

L122 – This sentence has little relevance to the argument presented and should be removed

L132 – An important section as it provides clear context to the ESM course and I thank the authors for including this within the paper.

L184 – The autoethnographic approach is an interesting approach to examine the course through the lens of the educators. Were student-views considered at all during the process? If not that may be an avenue to explore in the future.

L209-217 – This paragraph is unnecessary as it is repetition of what has been stated previously within the text at multiple occasions. The last sentence is the only relevant part and the remainder should be deleted.

Where the authors are presenting direct self reflection on the course, the text should be defined from the general discussion. For example L228-232 and L239-243. Use of quotations marks should be considered for this

L262 – Missing word ‘or’ before ‘when’

Discussion section – Even acknowledging the limited evidence available on extreme sports and extreme sports medicine, the findings need further critical interpretation. The authors should provide evidence to support how the course design favourably compares to standard medicine and sports medicine teaching and learning theory and delivery. For example, the statement on facilitating lifelong learning is positive but authors provide no interpretation of this concept – See work by Fischer (2000), Field (2001), Lewis-Fitzgerald (2007), or Marzo (2018)

L340 – This statement does not reflect the study and needs revising. The method state that the autoethnographic reflections of three authors were employed in the study and the results section presents statements of reflection from EA, GS, and LT. Additionally the conclusion provides an explanation of the ESM practitioner, whereas the autoethnography reflection is of the ESM course

L380 and L382 – Duplication of reference Pain and Pain (2005)

L385 – Inconsistency in reference and missing information

L391 – Missing page information

L392 and L393 – Missing information or potentially repetition of the same reference

L442 – Repetition of reference number 3 – Delete

Reviewer 3 Report

Dear Authors,

I attach my review to the message.

Kind regards,

Reviewer
